# Impact of an Inter-Professional Clinic on Pancreatic Cancer Outcomes: A Retrospective Cohort Study

Gordon Taylor Moffat [1], Zachary Coyne [1], Hamzeh Albaba [2], Kyaw Lwin Aung [3], Anna Dodd [1], Osvaldo Espin-Garcia [4], Shari Moura [1], Steven Gallinger [5,6], John Kim [7], Adriana Fraser [1], Shawn Hutchinson [1], Carol-Anne Moulton [6], Alice Wei [8], Ian McGilvray [6], Neesha Dhani [1], Raymond Jang [1], Elena Elimova [1], Malcolm Moore [1], Rebecca Prince [1] and Jennifer Knox [1,*]

1   Department of Medical Oncology and Hematology, Princess Margaret Cancer Centre, University of Toronto, Toronto, ON M5G 1X6, Canada
2   Department of Oncology, Jack Ady Cancer Centre, University of Alberta, Lethbridge, AB T1J 1W5, Canada
3   Livestrong Cancer Institutes and Dell Medical School, The University of Texas at Austin, Austin, TX 78712, USA
4   Department of Biostatistics, Princess Margaret Cancer Centre, University Health Network, Toronto, ON M5G 1X6, Canada
5   Lunenfeld-Tanenbaum Research Institute, Mount Sinai Hospital Joseph, Toronto, ON M5G 1X5, Canada
6   Toronto General Hospital, University Health Network, Toronto, ON M5G 2C4, Canada
7   Department of Radiation Medicine, Princess Margaret Cancer Centre, University of Toronto, Toronto, ON M5G 1X6, Canada
8   Department of Surgery, Memorial Sloan Kettering Cancer Center, Weill-Cornell School of Medicine, Cornell University, New York City, NY 10065, USA
*   Correspondence: jennifer.knox@uhn.ca; Tel.: +1-416-946-2399; Fax: +1-416-946-6546

**Abstract: Background:** Pancreatic ductal adenocarcinoma (PDAC) presents significant challenges in diagnosis, staging, and appropriate treatment. Furthermore, patients with PDAC often experience complex symptomatology and psychosocial implications that require multi-disciplinary and inter-professional supportive care management from health professionals. Despite these hurdles, the implementation of inter-professional clinic approaches showed promise in enhancing clinical outcomes. To assess the effectiveness of such an approach, we examined the impact of the Wallace McCain Centre for Pancreatic Cancer (WMCPC), an inter-professional clinic for patients with PDAC at the Princess Margaret Cancer Centre (PM). **Methods:** This retrospective cohort study included all patients diagnosed with PDAC who were seen at the PM before (July 2012–June 2014) and after (July 2014–June 2016) the establishment of the WMCPC. Standard therapies such as surgery, chemotherapy, and radiation therapy remained consistent across both time periods. The cohorts were compared in terms of survival rates, disease stage, referral patterns, time to treatment, symptoms, and the proportion of patients assessed and supported by nursing and allied health professionals. **Results:** A total of 993 patients were included in the review, comprising 482 patients pre-WMCPC and 511 patients post-WMCPC. In the multivariate analysis, adjusting for ECOG (Eastern Cooperative Oncology Group) and stage, it was found that post-WMCPC patients experienced longer median overall survival (mOS, HR 0.84, 95% CI 0.72–0.98, *p* = 0.023). Furthermore, the time from referral to initial consultation date decreased significantly from 13.4 to 8.8 days in the post-WMCPC cohort (*p* < 0.001), along with a reduction in the time from the first clinic appointment to biopsy (14 vs. 8 days, *p* = 0.022). Additionally, patient-reported well-being scores showed improvement in the post-WMCPC cohort (*p* = 0.02), and these patients were more frequently attended to by nursing and allied health professionals (*p* < 0.001). **Conclusions:** The implementation of an inter-professional clinic for patients diagnosed with PDAC led to improvements in overall survival, patient-reported well-being, time to initial assessment visit and pathological diagnosis, and symptom management. These findings advocate for the adoption of an inter-professional clinic model in the treatment of patients with PDAC.

**Keywords:** pancreatic cancer; inter-professional clinic; improved care; time to biopsy; overall survival

## 1. Introduction

Pancreatic ductal adenocarcinoma (PDAC) is the third leading cause of cancer-related death in Canada. In 2023, an estimated 7200 new cases of PDAC emerged, and 5900 died from the disease [1]. Despite treatment advances, the five-year median overall survival (mOS) rate of patients with PDAC remains low, at approximately 13% [2]. Upon diagnosis, the majority (80–85%) of patients with PDAC present with inoperable disease, defined as either locally advanced or metastatic, and a majority experience substantial disease-related symptom burden and a median survival of less than one year [3,4].

Systemic chemotherapy is the mainstay of treatment for patients with locally advanced or metastatic pancreatic cancer, yet the available treatment options remain limited [5]. For patients with a favorable performance status, combination chemotherapy is typically the preferred approach [3,4,6,7], while those with a poorer performance status often receive single-agent chemotherapy or best supportive care (BSC) [8]. Regrettably, the promising therapeutic benefit of immunotherapy and targeted therapies has not yet been translated into the management of PDAC. Consequently, there persists an urgent request for clinical trials aimed to explore novel treatment options in this patient population [5].

Managing patients with PDAC presents significant complexity, due to the prevalence of myriad physical and psychological symptoms, frequent oncologic complications of the disease, and the limited timeframes available for treatment intervention [9]. Emerging evidence supports the adoption of an inter-professional collaborative approach in patient care, resulting in a higher number of patients receiving multimodality anti-cancer therapy that includes surgery, radiotherapy, multi-agent chemotherapy, a perioperative approach, and combined chemo-radiotherapy, and enrolling on clinical trials, consequently improving survival rates [10,11]. Moreover, this approach ensures timely management of their symptoms, nutritional requirements, and social concerns, ultimately enhancing their quality of life [10–12].

In this study, we conducted a retrospective evaluation to assess the effects of an inter-professional approach to caring for pancreatic cancer patients referred and treated within the Wallace McCain Centre for Pancreatic Cancer (WMCPC) at Princess Margaret Cancer Centre (PM). Our hypothesis is that implementing an initial one-visit inter-professional clinic would lead to patients receiving the most appropriate treatment plan in a timely manner.

## 2. Methods

### 2.1. Setting, Participants, Study Design, and Data Source

The WMCPC was established at PM to deliver inter-professional care for patients and their families facing a diagnosis of PDAC. The vision of the WMCPC is to be a global leader in comprehensive pancreatic cancer care. The WMCPC is an integrated multidisciplinary and inter-professional program for pancreatic cancer patients, committed to providing high-quality clinical, academic, and research-focused care to improve outcomes throughout the cancer journey, with an overarching goal to reduce the burden of pancreatic cancer for patients and families at the provincial, national, and international level.

The inter-professional team at the WMCPC comprises a diverse range of specialists, including surgical oncologists, medical oncologists, radiation oncologists, radiologists, pathologists, clinical fellows, a dedicated clinical nurse specialist (CNS), specialized oncology nurses, an administrative assistant (AA), clinical trial nurses, a genetic counselor, a research program manager, research coordinators, a registered dietician, and a social worker. Each member, already engaged in caring for patients with pancreatic cancer within their respective roles, agreed to participate in a dedicated weekly inter-professional new patient clinic to allow these specialties to assess new patients promptly. Significant delays in care, spanning from the receipt of referral to diagnosis and the initiation of treatment and supportive care, were recognized as primary concerns within this patient population. Thus, the WMCPC was established to serve these concerns.

To facilitate prompt processing of referrals, the WMCPC assigned a dedicated CNS and an AA to initiate and coordinator care requirements. Referrals are rapidly triaged to

determine which oncology specialties should assess the patient at the initial visit. The majority of referrals to the WMCPC request expedited diagnostic workup and/or assessment for available clinical trials, either upon initial diagnosis or at the time of disease progression. Unlike standard oncology referrals to PM and other cancer centers, the WMCPC evaluates patients with suspected pancreatic cancer even in the absence of a confirmed tissue diagnosis. This acknowledges the challenge of arranging timely diagnostic biopsies outside of oncology specialties.

Additionally, the radiographic imaging of each new patient is formally reviewed at a multidisciplinary cancer conference (MCC), which is attended by the inter-professional team of the WMCPC. During the MCC, the team reviews radiographic imaging, prioritizes the best method to obtain tissue diagnosis, discusses treatment plans, and ensures that coordinated care plans are in place for each patient. Since fully adopting this inter-professional care approach in July 2014, the WMCPC has assessed approximately 3780 patients.

This retrospective study included all patients (*n* = 993) diagnosed with PDAC and treated at PM from July 2012 to July 2016. The WMCPC was fully operational in July 2014; consequently, the cohorts for this comparative study included patients seen for a new patient appointment pre-WMCPC (July 2012 to June 2014) and post-WMCPC (July 2014 to June 2016). The primary outcome of measure was mOS, and the secondary outcomes included various aspects of patient-related wait times and clinical management. Demographic details, disease characteristics, treatment plans, and clinical outcomes were abstracted from electronic medical records (EMR) into the WMCPC REDCap (Research Electronic Data Capture) Database. REDCap Database is a secure data management system housed on the internal server of our institution and only accessed by encrypted computer by users with a REDCap account and approval by the database managers. Standard treatments included neo-adjuvant/peri-operative chemotherapy with/without radiotherapy, surgery, adjuvant chemotherapy, palliative chemotherapy, and palliative radiotherapy. Cohorts were compared for mOS, stage at diagnosis, Eastern Cooperative Oncology Group Performance Status (ECOG PS), time from referral to initial first clinic assessment, time to first treatment, Edmonton Symptom Assessment Score (ESAS) for pain, tiredness, and general well-being in the first 6 months of treatment (scores were averaged due to significant missing data and scores of $\geq 4$ were considered significant), participation in clinical and translational trials and proportion of patients seen by the CNS, registered dietician, and/or social worker. This study was approved by the University Health Network (UHN) Research Ethics Board.

*2.2. Statistical Analysis*

For comparison of outcome measures pre- and post-WMCPC, a Mann–Whitney U test or a t-test was used for continuous variables. A chi-squares test or a Fisher's exact test was used for nominal/categorical variables. The log-rank test and the Kaplan–Meier method were used to compare OS between patients seen pre- and post-WMCPC in a univariate and multivariate analysis. Overall survival was defined as time from date of diagnosis until death. In addition, hazard ratios (HRs), adjusted hazard ratios (aHRs) and their corresponding 95% confidence intervals (CIs) were calculated using the Cox proportional hazards model. For aHRs, ECOG and stage were used as covariates.

**3. Results**

*3.1. Patient Characteristics and Demographics*

A total of 993 patients with PDAC were included (Table 1), 482 patients in the pre-WMCPC cohort and 511 patients in the post-WMCPC cohort. Age was similar between the two cohorts (median 67.4 years, range 27.5–95.6), with more male patients (54%) overall. At the first appointment, physicians assessed ECOG PS in 29% of patients in the pre-WMCPC cohort versus 54% of patients in post-WMCPC cohort. Twenty-three percent of patients in the pre-WMCPC cohort had an ECOG PS status of 0–1 versus 34% in the post-WMCPC cohort. Patients with a ECOG PS of $\geq 2$ doubled in the post- WMCPC cohort (6% vs.

12%). Cancer stage at the time of diagnosis was comparable between the two cohorts for borderline resectable pancreatic cancer (11% vs. 10%) and advanced disease (50% vs. 55%); however, there were more localized and resectable cancers in the post-WMCPC cohort (17% in pre-WMCPC vs. 24%, *p* = 0.0069). Most patients (85–90%) had ductal adenocarcinoma histology; other histology included adeno-squamous cancer. More patients were referred without a pathological diagnosis in the post-WMCPC cohort than pre-WMCPC cohort (49% vs. 39%). Overall, 73% of patients with PDAC seen at PM for a new patient appointment remained at PM for their cancer care and treatment, including palliative care.

**Table 1.** Characteristics of patients with PDAC assessed at PM.

| Covariate | Total | Pre-WMCPC | Post-WMCPC | *p*-Value |
|---|---|---|---|---|
| | (*n* = 993) | (*n* = 482) | (*n* = 511) | |
| **Age** | | | | |
| Median (Range) | 67.4 (27.5, 96.6) | 67.6 (27.5, 95.7) | 67.3 (29.2, 96.6) | 0.62 |
| Unknown | 193 | 145 | 48 | |
| **Sex** (*N*, %) | | | | |
| Male | 538 (54) | 262 (54) | 276 (54) | 0.97 |
| Female | 454 (46) | 220 (46) | 234 (46) | |
| Unknown | 1 (0) | 0 | 1 (0) | |
| **ECOG PS** (*N*, %) | | | | |
| 0–1 | 286 (29) | 112 (23) | 174 (34) | **<0.001** |
| ≥2 | 91 (9) | 29 (6) | 62 (12) | |
| Unknown | 616 (62) | 341 (71) | 275 (54) | |
| **Stage** (*N*, %) | | | | |
| I–IIA | 206 (21) | 83 (17) | 123 (24) | **<0.001** |
| IIB | 106 (11) | 53 (11) | 53 (10) | |
| III–IV | 520 (53) | 240 (50) | 280 (55) | |
| Unknown | 161 (16) | 106 (22) | 55 (11) | |
| **Histology** (*N*, %) | | | | |
| Adenocarcinoma | 867 (87) | 409 (85) | 458 (90) | **0.0098** |
| Adeno-squamous | 13 (1) | 4 (1) | 9 (2) | |
| Unknown | 113 (11) | 69 (14) | 44 (9) | |
| **Treated at PM** (*N*, %) | | | | |
| Yes | 725 (73) | 346 (72) | 379 (74) | 0.43 |
| No | 268 (27) | 136 (28) | 132 (26) | |
| **Type of first treatment** (N, %) | **(*n* =584)** | **(*n* = 293)** | **(*n* = 291)** | |
| Curative surgery | 149 (26) | 85 (29) | 64 (22) | |
| Neoadjuvant chemotherapy | 27 (5) | 8 (3) | 19 (7) | |
| Neoadjuvant chemo-radiotherapy | 6 (1) | 3 (1) | 3 (1) | |
| Adjuvant chemotherapy | 18 (3) | 5 (2) | 13 (4) | |
| Palliative chemotherapy | 300 (51) | 153 (52) | 147 (51) | |
| Palliative chemo-radiotherapy | 22 (4) | 18 (6) | 4 (1) | |
| Palliative surgery | 12 (2) | 7 (2) | 5 (2) | |
| Best supportive care | 42 (7) | 12 (4) | 30 (10) | |
| Other | 6 (1) | 2 (1) | 4 (1) | |
| Unknown | 2 (0) | 0 (0) | 2 (1) | |
| **Surgery received** (*N*, %) | | | | |
| Curative | 264 (27) | 114 (24) | 150 (29) | |
| Palliative bypass | 36 (4) | 24 (5) | 12 (2) | **0.037** |
| None | 639 (64) | 315 (65) | 324 (63) | |
| Unknown | 54 (5) | 29 (6) | 25 (5) | |

WMCPC: Wallace McCain Centre for Pancreatic Cancer; ECOG PS: Eastern Cooperative Oncology Group Performance Status; PM: Princess Margaret Cancer Centre. **The two groups were balanced.**

### 3.2. Primary Outcome Analysis

In the univariate Kaplan–Meier analysis, there was a trend towards a longer mOS in the post-WMCPC cohort compared to the pre-WMCPC cohort (10.9 vs. 9.6 months, *p* = 0.055), as seen in Figure 1. The multivariate Cox regression for OS adjusting for ECOG performance status and stage at the time of diagnosis revealed a significant hazard ratio for post-WMCPC versus pre-WMCPC of 0.84 (95% CI 0.72–0.98, *p* = 0.023). This significant difference persisted when treatment intent was included in the multivariate analysis (HR 0.81, 95% CI 0.7–0.94, *p* = 0.007), but not when tumor type and surgery were

included, although the effect direction remained (HR 0.92); the *p*-value for tumor type was non-significant, while the *p*-value for surgery was highly significant. Further analysis showed that patients undergoing curative-intent surgery in the post-WMCPC cohort had better survival.

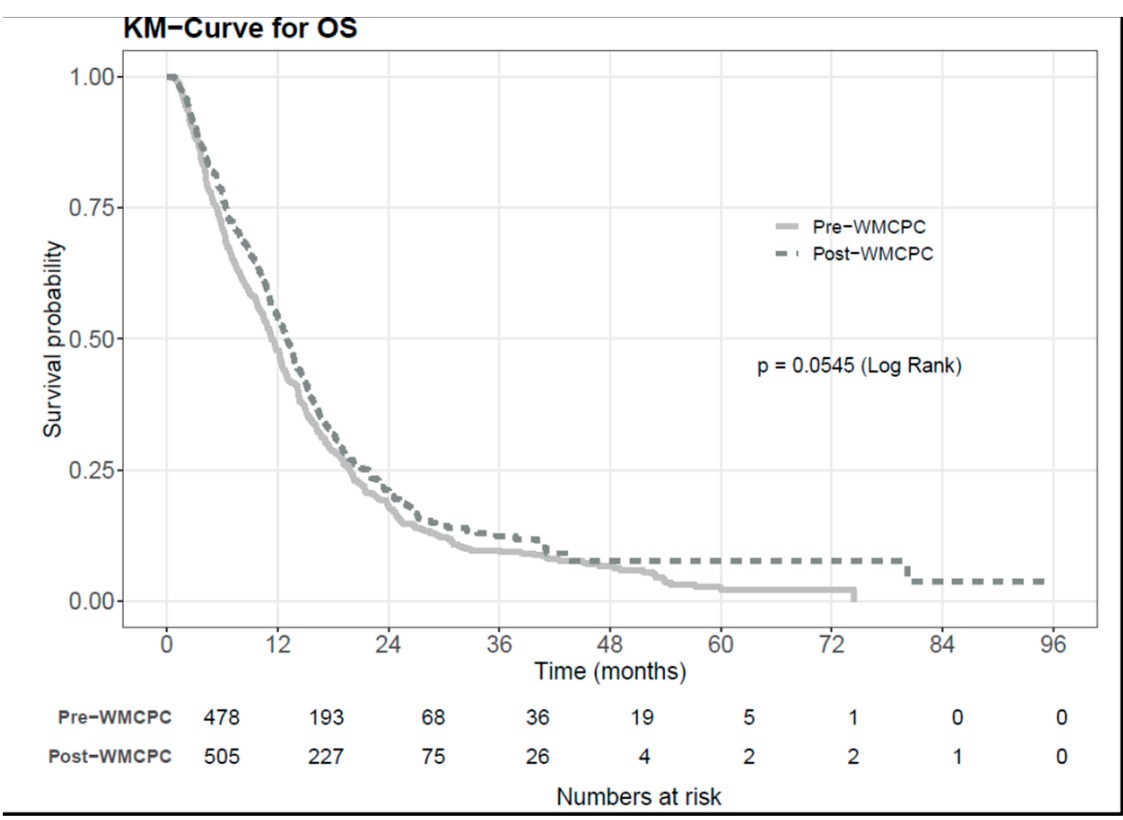

**Figure 1.** Kaplan–Meier curve for overall survival.

### 3.3. Secondary Outcome Analysis

The time from the receipt of referral to the first clinic appointment was 10 versus 7 days in the pre- versus post-WMCPC cohorts ($p < 0.001$) (Table 2). In patients without a diagnostic biopsy, the time from the initial new patient appointment to the diagnostic biopsy improved from 14 days in the pre-WMCPC cohort to 8 days in the post-WMCPC cohort ($p = 0.02$). The time from the initial new patient appointment to the initiation of first oncology treatment was similar between the two cohorts, which was 26 versus 23 days in pre- and post-WMCPC, respectively. There were no significant differences in ESAS pain and tiredness scores, but there was a significant difference in general well-being scores (3.7 vs. 4, $p = 0.02$) between the pre- and post-WMCPC cohort patients. Clinically significant ESAS scores (score $\geq$ 4) occurred in 28–65% of PDAC patients. Clinical trial participation significantly increased in the post-WMCPC cohort (8% vs. 12%, $p = 0.035$). For patients treated at the PM, there was a statistically significant increase in the proportion of patients assessed and supported by the CNS (31% pre-WMCPC vs. 50% post-WMCPC, $p < 0.001$), the dietician (9% pre-WMCPC vs. 35% post-WMCPC, $p < 0.001$), and social work (8% pre-WMCPCvs. 38% post-WMCPC, $p < 0.001$) in the post-WMCPC cohort.

**Table 2.** Secondary outcomes analysis.

| Outcomes | Total | Pre-WMCPC (*n* = 482) | Post-WMCPC (*n* = 511) | *p*-Value |
|---|---|---|---|---|
| Time from referral to first clinic visit (days) Median (minimum, maximum) Unknown | (*n* = 993) 8 (0, 374) 116 | 10 (0, 217) 114 | 8.8 (0, 374) | **<0.001** |
| Time from first visit to biopsy (days) Median (range) Unknown | (*n* =580) 12 (0, 567) 161 | 14 (0, 567) 88 | 8 (0, 245) 73 | **0.022** |
| Time to curative surgery Median (range) Unknown | (*n* = 149) 28 (5, 749) 41 | (*n* = 85) 34 (5, 145) 39 | (*n* = 64) 26 (6, 749) 2 | 0.48 |
| Time to neoadjuvant chemotherapy/ chemo-radiotherapy (days) Median (range) Unknown | (*n* = 33) 34 (16, 208) 1 | (*n* = 11) 33 (16, 121) 1 | (*n* = 22) 34 (17, 208) 0 | 0.93 |
| Time to adjuvant chemotherapy (days) Median (range) | (n = 18) 45 (23, 108) | (n = 5) 44 (35, 45) | (n = 13) 51 (23, 108) | 0.25 |
| Time to palliative chemotherapy (days) Median (range) Unknown | (*n* = 300) 34 (11, 670) 58 | (*n* = 153) 35 (11, 605) 22 | (*n* = 147) 32 (11, 670) | 0.3 |
| Time to palliative chemo-radiotherapy (days) Median (range) Unknown | (*n* = 22) 31.5 (6, 889) 4 | (*n* = 18) 31.5 (6, 97) 4 | (*n* = 4) 30.5 (17, 889) 0 | 0.074 |
| Time from first clinic to first oncology treatment at PM (days) Median (range) Unknown | (*n* = 583) 25 (0, 886) 84 | (*n* = 292) 26 (0, 721) 14 | (*n* = 291) 23 (1, 886) 70 | 0.14 |

WMCPC: Wallace McCain Centre for Pancreatic Cancer; ECOG PS: Eastern Cooperative Oncology Group Performance Status; PM: Princess Margaret Cancer Centre. Note: Treatment timing may vary significantly due to the complex nature of the disease and the high likelihood of complications, which can lead to delays in initiating treatments. **An improvement in time to first assessment and biopsy were demonstrated.**

## 4. Discussion

The implementation of an initial visit inter-professional clinic for patients with PDAC demonstrated remarkable improvements across various facets of patient care. In our study, we observed enhancements in patient outcomes such as mOS, well-being scores, increased participation in clinical trials, and greater involvement of nursing and allied health professionals. Additionally, notable reductions were seen in waiting times from referral to first appointment and diagnostic biopsy, addressing significant barriers to prompt treatment initiation.

The parallels drawn between our findings and those of previous studies underscore the effectiveness of inter-professional clinics in improving patient outcomes in PDAC. Studies by Hoehn et al. and Pawlik et al. demonstrated similar improvements in mOS, likely attributable to the facilitated cooperation and collaboration among oncologic specialties, the development of consensus-based recommendations and comprehensive management plans, and standardized treatment decisions facilitated by this model of care [11,13]. Similarly, the findings that an inter-professional clinic for patients with PDAC resulted in an increased use of multimodality therapy and clinical trial enrollment were also seen by Schiffman et al., along with a higher rate of patients that received neoadjuvant therapy and treatment overall [10]. Lastly, our study showed that an inter-professional clinic for patients with PDAC reduced the time from referral to first appointment, as in a study by Vaghjiani et al. [14]; however, we were also able to show a reduced time to biopsy, which can be a common barrier in care and can delay the prompt initiation of treatment, a critical factor influencing survival outcomes in these patients [9].

A significant contribution of our study is the documented improvement in patient well-being scores using an inter-professional clinic model. Further, we demonstrated a statistically significant increase in the frequency of services provided by CNS and allied health professionals at the WMCPC that can translate into optimal symptom management and re-

duced anxiety, stress, and burnout experienced by patients and their caregivers throughout the treatment course. Enhanced pain and symptom management not only improve mood and quality of life but also influence end-of-life care and extend survival [15,16]. Given the complex and clustered symptoms often seen in patients with PDAC, which require multi-modality treatment and toxicity-specific management, the integration of inter-professional clinics into oncology practice is warranted [10–12,17–19].

The efficiency of the single-visit inter-professional clinic model, as demonstrated in our study, stems from its capacity to streamline patient care through the consolidation of oncology specialties and the addition of dedicated CNS and allied health professionals for timely and comprehensive care. The WMCPC clinic model proves particularly effective in enabling the rapid triage of patient referrals and facilitating expedited diagnostic workup. It achieves this by reserving imaging and biopsy appointments/slots and, where possible, coordinating follow-up appointments. This model eliminates the time delays, patient costs (parking, fuel, taxis, ride share, time off work, etc.), and associated anxiety, as patients were scheduled and seen in each specialty's clinic independently allowing for the real-time discussion of each patient's case in person, followed by a MCC, which has also been shown to improve patient-related outcomes [20], and provides a definitive management plan at the time of discharge from clinic the same day, further reducing the ambiguity, anxiety, and distress experienced by patients and their families. This study represents the first two years of an inter-professional care model, where many pathways and efficiencies were still being optimized in the context and limits of the PM and interaction with the surrounding medical community, yet the improvements were seen early. A follow-up of similar benchmarks will be assessed now in the post-pandemic era to allow further assessment of progress. Further, studies have shown that this inter-professional model of care can standardize practices and eliminate socioeconomic disparities in this patient population [21]. Acknowledging that challenges exist in establishing an inter-professional clinic, including the need for dedicated institutional resources such as financial allocation, clinic space availability, skilled and experienced health care providers, and time commitment from team members [11], we nonetheless advocate for its pivotal role and merit further consideration. This study adds to a growing body of literature on this topic and the care of patients with PDAC.

*Limitations*

This study has limitations. First, it is a single-center retrospective review of patients with PDAC; thus, patient selection bias may have occurred. Second, although all patients were initially seen and assessed at PM, only two-thirds were treated at PM, resulting in missing data for patients treated at other regional or community cancer centers. Most survival data were available; however, these missing data could have affected comparisons and conclusions. Third, while our clinic is comprehensive, we recognize the clinic operates within an existing institutional system with barriers and delays that are out of our control, which include access to endoscopic procedures, chemotherapy booking times, and operating room schedules that may impact our results. Notwithstanding, our results demonstrated a significant improvement in overall survival, which might increase if the system-based delay processes are enhanced.

Of note, the study conducted by Conroy et al., demonstrating the efficacy of FOLFIRI-NOX (leucovorin (folinic acid), fluorouracil, irinotecan, and oxaliplatin.), was published and approved for funding in Canada in 2011, whereas the study examining nab-Paclitaxel plus Gemcitabine by von Hoff et al. was published in 2013 and received funding approval in Canada in 2015. The disparity in approval timelines and usage patterns may have influenced the analysis of overall survival between cohorts. However, a current study with updated data is underway, which will address this potential factor as both regimens were approved and utilized during this included study period.

## 5. Conclusions

The implementation of a streamlined inter-professional clinic for the care of patients with PDAC improved overall survival, patient well-being scores, waiting times for appointments and diagnostic biopsies, and the use of CNS and allied health professionals for symptom and supportive care management. These findings have significant implications for planning future care delivery models for pancreatic cancer patients and demonstrate the value of continuing such interventions to achieve superior health outcomes for patients with PDAC.

**Author Contributions:** G.T.M.: investigation, data curation, writing—original draft preparation, writing—review and editing, and visualization. Z.C.: writing—review and editing. H.A.: methodology, formal analysis, investigation, data curation, writing—original draft preparation, and writing—review and editing. K.L.A.: investigation and writing—review and editing. A.D.: resources, writing—review and editing, and project administration. O.E.-G.: methodology, software, validation, formal analysis, and writing—review and editing. S.M.: writing—review and editing. S.G.: conceptualization, resources, writing—review and editing, and funding acquisition. J.K. (John Kim): writing—review and editing. A.F.: writing—review and editing. S.H.: data curation. C.-A.M.: writing—review and editing. A.W.: writing—review and editing. I.M.: writing—review and editing. N.D.: writing—review and editing. R.J.: writing—review and editing. E.E.: writing—review and editing. M.M.: conceptualization, resources, writing—review and editing, and funding acquisition. R.P.: methodology, data curation, writing—review and editing, and visualization. J.K. (Jennifer Knox): conceptualization, resources, writing—review and editing, supervision, and funding acquisition. All authors have read and agreed to the published version of the manuscript.

**Funding:** The Princess Margaret Cancer Foundation Wallace McCain Centre for Pancreatic Cancer (WMCPC).

**Institutional Review Board Statement:** This study was approved by the University Health Network (UHN) Research Ethics Board (16-6279, initial approval 13-January-2017, re-approval 03-January-2024; 18-5155, initial approval 15-June-2018, re-approval 12-July-2023).

**Informed Consent Statement:** Informed consent was obtained from all subjects involved in the study.

**Data Availability Statement:** The data presented in this study are available on request from the corresponding author.

**Conflicts of Interest:** A.W.: Histosonics—consulting, Ipsen—clinical trial funding. Jennifer J. Knox: consultant for AZ, Merck, Roche, Eisai, Ibsen, and Pfizer; grant/research support from AZ, Ibsen, Merck, and Roche. The other authors declare no conflicts of interest.

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
