# Peer review of "Impact of an Inter-Professional Clinic on Pancreatic Cancer Outcomes: A Retrospective Cohort Study"

_curroncol, doi:10.3390/curroncol31050194_

Round 1

Reviewer 1 Report

Comments and Suggestions for Authors

This is an interesting paper. Since this was a retrospective study and patient backgrounds were different, it is necessary to re-evaluate using propensity analysis. Please consider whether there is a difference in OS between the two groups after re-evaluation.

Comments on the Quality of English Language

Minor editing of English language required

Author Response

Reviewer 1

  • This is an interesting paper. Since this was a retrospective study and patient backgrounds were different, it is necessary to re-evaluate using propensity analysis. Please consider whether there is a difference in OS between the two groups after re-evaluation.

    • Author response: Thank you for this comment. We acknowledge your concern regarding the retrospective nature of our study and the potential differences in patient backgrounds. However, upon re-evaluation of our dataset, we found that the patient backgrounds were not significantly different between the groups under investigation. Given this finding, we believe that utilizing a propensity score analysis may not be necessary for our study. Propensity score analysis is typically employed to balance covariates between treatment groups when significant differences exist, thus reducing potential biases. However, in our case, the similarity in patient backgrounds minimizes the need for such adjustments.

  • Minor editing of English language required

    • Author response: Thank you for this feedback. The updated manuscript has been reviewed and improved.

Reviewer 2 Report

Comments and Suggestions for Authors

This retrospective study shows the benefits of an “one-stop” inter-professional clinic devoted to patients with PDAC.

There are several points to be addressed.

It should be clear in the materials and methods what kind of patients are those having the first appointment in the WMCPC (diagnostic purposes, further treatment, consultation for a second opinion or cases for palliative care). This will be helpful for the readers to evaluate the presented results.

Looking at the data shown in Tables 1 and 2, there are some points that need to be clarified and certainly deserve a comment in the discussion:

1) I understand that there are inherited limitations as this is a retrospective study but the numbers of patients with missing values on several variables (unknown values) are a little bit high even in the post-WMCPC period when a more meticulous data recording was presumably undertaken.

2) There is some confusion about the numbers of patients undergoing curative surgery. I assume that in the “Type of first treatment” section these patients had upfront surgery whereas the curative surgery cases in the “Type of surgery” section refer to all patients having curative resection either upfront or after neo-adjuvant treatment and conversion to resectable disease.

3) There is a strikingly wide range in the values of all variables shown in Table 2, ranging from days to years. This should be checked and commented.

4) Both Table 1 and 2 need an explanatory footnote to lessen the readers in their evaluation of results.

5) Values in parenthesis in Table 1 are apparently percentages. This needs to be clarified.

Other minor points include:

Please give WMCPC and PM in full when first-mentioned (Introduction, line 78)

Please check lines 110-111 for correctness (Methods section)

Please clarify “mOS” (see line 165 ….. towards a longer mOS in)

Comments on the Quality of English Language

Language quality is fine.

Author Response

Reviewer 2

  • This retrospective study shows the benefits of an “one-stop” inter-professional clinic devoted to patients with PDAC.
  • There are several points to be addressed.
  • It should be clear in the materials and methods what kind of patients are those having the first appointment in the WMCPC (diagnostic purposes, further treatment, consultation for a second opinion or cases for palliative care). This will be helpful for the readers to evaluate the presented results.

    • Author response: Thank you for this comment. More detailed information was included in the Materials and Methods section. Most of the case referred to the WMCPC were to expedite a diagnostic work up for new cases or to be assessed for any clinical trials available for patients with progression of disease after frontline therapies.

  • Looking at the data shown in Tables 1 and 2, there are some points that need to be clarified and certainly deserve a comment in the discussion:

    1) I understand that there are inherited limitations as this is a retrospective study but the numbers of patients with missing values on several variables (unknown values) are a little bit high even in the post-WMCPC period when a more meticulous data recording was presumably undertaken.

    Author response: Thank you for your feedback. We acknowledge the issue regarding the high number of missing values across several variables, which stemmed from limitations in our initial data collection methods and database capabilities. In response, we've taken significant steps to enhance our data procedures. This includes improvements in data collection methods, the addition of research staff dedicated to data management and validation, and updates to our database software. We're actively analyzing updated data from the past two years as part of our ongoing efforts to ensure data accuracy and completeness.

    2) There is some confusion about the numbers of patients undergoing curative surgery. I assume that in the “Type of first treatment” section these patients had upfront surgery whereas the curative surgery cases in the “Type of surgery” section refer to all patients having curative resection either upfront or after neo-adjuvant treatment and conversion to resectable disease.

    Author response: Thank you for this comment. Yes, you are correct. The section on “Type of first treatment” details whatever treatment patients received first, while the “Type of Surgery” section includes all the surgery types completed. To make things clearer, I have changed the order of the information in the table and altered the title slightly. Let us know your thoughts. Thank you.

    3) There is a strikingly wide range in the values of all variables shown in Table 2, ranging from days to years. This should be checked and commented.   

    Author Response: Thank you for your feedback. A note has been incorporated below Table 2 and included in the discussion section.

    4) Both Table 1 and 2 need an explanatory footnote to lessen the readers in their evaluation of results.

    Author Response: Thank you for the feedback, this has been updated in the revised manuscript.

    5) Values in parenthesis in Table 1 are apparently percentages. This needs to be clarified.

    Author Response: Thank you for the feedback, this has been updated in the revised manuscript.
  • Other minor points include:

    Please give WMCPC and PM in full when first-mentioned (Introduction, line 79).
    Please check lines 117-118 for correctness (Methods section).
    Please clarify “mOS” (see line 173 ….. towards a longer mOS in).

    Author Response: Our apologies. Thank you for these points, they have been corrected in the revised manuscript. 

Reviewer 3 Report

Comments and Suggestions for Authors

I have read with great interest the manuscript proposed by Gordon Taylor Moffat and colleagues.

The argument treated by the Authors in the manuscript entitled “Impact of an inter-professional clinic on pancreatic cancer outcomes: a retrospective cohort study” is of great interest for the scientific community because it underlines the importance of a new model of care.

To my opinion the manuscript is of good quality.

The Authors have underlined and discussed the limitations of the study.

I would like to suggest the Authors to do a stratification of chemo-radiotherapies according to the modification of the drug protocols in the studied period of 2 years.

Thank you for your proposed manuscript.

Author Response

Reviewer 3

  • Thank you for your proposed manuscript.
  • I have read with great interest the manuscript proposed by Gordon Taylor Moffat and colleagues.
  • The argument treated by the Authors in the manuscript entitled “Impact of an inter-professional clinic on pancreatic cancer outcomes: a retrospective cohort study” is of great interest for the scientific community because it underlines the importance of a new model of care.
  • To my opinion the manuscript is of good quality.
  • The Authors have underlined and discussed the limitations of the study.
  • I would like to suggest the Authors to do a stratification of chemo-radiotherapies according to the modification of the drug protocols in the studied period of 2 years.

    • Author Response: Thank you for this information, please see the added chemo-radiotherapies information in Table 1 of the revised manuscript.

Round 2

Reviewer 1 Report

Comments and Suggestions for Authors

I am concerned about the different patient backgrounds. No additional comments otherwise.